# PlanGAN: Model-based Planning With Sparse Rewards and Multiple Goals

**Henry Charlesworth**     **Giovanni Montana**
Warwick Manufacturing Group
University of Warwick
Coventry, United Kingdom
{H.Charlesworth, G.Montana}@warwick.ac.uk

## Abstract

Learning with sparse rewards remains a significant challenge in reinforcement learning (RL), especially when the aim is to train a policy capable of achieving multiple different goals. To date, the most successful approaches for dealing with multi-goal, sparse reward environments have been model-free RL algorithms. In this work we propose PlanGAN, a model-based algorithm specifically designed for solving multi-goal tasks in environments with sparse rewards. Our method builds on the fact that any trajectory of experience collected by an agent contains useful information about how to achieve the goals observed during that trajectory. We use this to train an ensemble of conditional generative models (GANs) to generate plausible trajectories that lead the agent from its current state towards a specified goal. We then combine these imagined trajectories into a novel planning algorithm in order to achieve the desired goal as efficiently as possible. The performance of PlanGAN has been tested on a number of robotic navigation/manipulation tasks in comparison with a range of model-free reinforcement learning baselines, including Hindsight Experience Replay. Our studies indicate that PlanGAN can achieve comparable performance whilst being around 4-8 times more sample efficient.

## 1   Introduction

One of the primary appeals of reinforcement learning (RL) is that it provides a framework for the autonomous learning of complex behaviours without the need for human supervision. In recent years RL has had significant success in areas such as playing video games [1, 2], board games [3, 4] and robotic control tasks [5, 6, 7]. Despite this, progress in applying RL to more practically useful environments has been somewhat limited. One of the main problems is that RL algorithms generally require a well-shaped, dense reward function in order to make learning progress. Often a reward function that fully captures the desired behaviour of an agent is not readily available and has to be engineered manually for each task, requiring a lot of time and domain-specific knowledge. This defeats the point of designing an agent that is capable of learning autonomously. A more general approach is to learn with sparse rewards, where an agent only receives a reward once a task has been completed. This is much easier to specify and is applicable to a wide range of problems, however training becomes significantly more challenging since the agent only receives infrequent feedback at the end of every rollout. This becomes especially challenging in the case of goal-conditioned RL [8, 9], where the aim is to train a policy that can achieve a variety of different goals within the environment.

Much of RL's success has come with model-free approaches, where the policy is learned directly from the reward signal obtained by interacting with the environment. However recently there has been a lot of interest in applying model-based approaches to the same kind of problems [7, 10, 11]. One

of the main drawbacks of model-free RL algorithms is that they tend to be very sample inefficient, requiring a huge number of interactions with the environment in order to make learning progress. On the other hand, model-based methods make use of a learned model to plan their actions without directly interacting with the environment. Learning a model allows these methods to make use of a lot more information that is present in the observed transitions than just the scalar reward signal, and so generally this leads to a significant improvement in sample efficiency. This efficiency can sometimes come at the cost of worse asymptotic performance due to errors in the model introducing a bias towards non-optimal actions, although current state of the art approaches [7, 10] are able to achieve comparable performance to some of the best model-free approaches [12, 13]. However, as with most RL algorithms, model-based approaches generally need a dense reward signal to work well. We are not aware of a model-based approach specifically designed to work in the sparse-reward, multi-goal setting.

To date, the most successful general-purpose RL algorithm for dealing with sparse rewards and multiple goals is Hindsight Experience Replay (HER) [8], a model-free algorithm. HER works by taking advantage of the fact that, when learning a goal-conditioned policy with an off-policy RL algorithm, observed transitions from a trajectory can be re-used as examples for attempting to achieve *any* goal. In particular, by re-labelling transitions with goals achieved at a later point during the same trajectory HER trains the goal-conditioned policy on examples that actually led to success — hence obtaining a much stronger learning signal.

In this paper we present PlanGAN, a model-based algorithm that can naturally be applied to sparse-reward environments with multiple goals. The core of our method builds upon the same principle that underlies HER — namely that any goal observed during a given trajectory can be used as an example of how to achieve that goal from states that occurred earlier on in that same trajectory. However, unlike HER, we do not directly learn a goal-conditioned policy/value function but rather train an ensemble of Generative Adversarial Networks (GANs) [14] which learn to generate plausible future trajectories *conditioned on achieving a particular goal*. We combine these imagined trajectories into a novel planning algorithm that can reach those goals in an efficient manner.

We test PlanGAN on a number of robotic manipulation and navigation tasks and show that it can achieve similar levels of performance to leading model-free methods (including Hindsight Experience Replay) but with substantially improved sample efficiency. The primary contribution of this paper is to introduce the first model-based method which is explicitly designed for multi-goal, sparse reward environments, leading to a significant improvement in sample efficiency.

## 2   Related Work

A number of model-based approaches have utilised explicit planning algorithms, but have mostly been applied to single tasks with relatively dense rewards. Nagabandi et al. [15] use iterative random shooting within a deterministic neural network dynamics model in order to solve a number of continuous control tasks. Hafner et al. [16] learn a latent representation from images and then plan within this latent space using CEM. Nagabandi et al. [17] use a similar planning algorithm (MPPI) [18] within an ensemble of learned models in order to perform dexterous manipulation tasks.

Other methods have had success with a hybrid approach, combining elements of model-based and model-free RL, and as in this work often use ensembles of models in order to improve robustness. STEVE [19] uses rollouts produced by an ensemble of models and Q-functions in order to obtain a robust estimate for the Q-learning target. Model-Ensemble TRPO [20] uses an ensemble of models as a simulator for running a model-free RL algorithm (trust-region policy optimisation) whilst maintaining some level of uncertainty for when the model's predictions are valid. I2A [21] learns to interpret imagined trajectories generated by a model to augment the model-free training of a policy/value function. Temporal Difference Models (TDMs) [22] try to link model-based and model-free RL in the context of time-dependent, goal-conditioned value functions. Here, the model is itself the goal-conditioned value function, and is learned with model-free, off-policy RL. However, they require a meaningful distance metric between states to be defined and so do not work with fully sparse rewards. Nasiriany et al. [23] combine TDMs as an implicit model with a planning algorithm that allows them to plan over multiple abstract sub-goals. They apply this to solve long-horizon, goal-conditioned tasks directly from images.

Azizzadenesheli et al. [24] use a Wasserstein GAN with spectral normalisation to learn a predictive model that they use with Monte-Carlo Tree Search to solve ATARI games. Although they do not find particularly strong results overall, they show that they are able to learn an extremely accurate model with stable training of the GAN even in a non-stationary environment. A significant difference with our work is that they train a GAN that takes an action and a state and predicts the next state, whereas we train the GANs to imagine full trajectories (also their focus is on image-based environments). GANs have also been used for curriculum learning in goal-conditioned RL [25], where a generator was trained to propose goals at an appropriate level of difficulty for the current agent to achieve.

In terms of learning with sparse rewards, a number of approaches have had success by providing the agent with intrinsic rewards in order to aid with exploration [26, 27, 28]. However, in the multi-goal setting a majority of the most successful approaches have built upon Hindsight Experience Replay (HER) [8]. Zhao & Tresp [29] improve HER's performance on certain robotics environments by more frequently resampling trajectories where the objects have higher energy. Fang et al. [30] propose an adaptive mechanism to select failed experiences based on a combination of the diversity of the achieved goals and their proximity to the desired goals. Liu et al. [31] propose a complementary re-labelling scheme in the context of a competitive exploration game between two agents in order to supplement HER. He at al. [32] introduce a method that combines HER with maximum entropy RL.

Taking a different approach (but still closely related to HER), Ghosh et al. [33] introduce a method that learns goal-conditioned policies without explicitly using reinforcement learning. They use supervised behavioural cloning (a form of imitation learning) to train a policy to reach the goals that have been observed on the trajectories the agent itself has generated. Whilst simpler than HER, it does not use a model and does not claim to significantly improve upon HER's sample efficiency.

## 3 Preliminaries

### 3.1 Goal-Conditioned Reinforcement Learning

We consider the problem of an agent interacting within an environment in order to learn how to achieve any given goal $g$ from a set of possible goals $\mathcal{G}$. We assume that the environment is fully observable and can be described by: a set of states, $\mathcal{S}$; a set of possible actions, $\mathcal{A}$; a distribution of initial states, $p(s_0)$; and a transition function $P(s_{t+1}|s_t, a_t)$ $(s_t, s_{t+1} \in \mathcal{S}, a_t \in \mathcal{A})$. In the standard reinforcement setting we have a reward function, $R(s_t, a_t, s_{t+1})$. In the goal-conditioned setting the reward also depends on the goal that the agent is trying to achieve, i.e. $R(s_t, a_t, s_{t+1}, g)$. Assuming that goals are sampled from some distribution $p(\mathcal{G})$, the aim of goal-conditioned RL is to learn a policy, $\pi(s_t, g)$, that maximises the expected discounted sum of future rewards:

$$\mathbb{E}_{\substack{s_0 \sim p(s_0) \\ g \sim p(\mathcal{G}) \\ a_t \sim \pi(s_t, g) \\ s_{t+1} \sim P(s_{t+1}|s_t, a_t)}} \left[ \sum_{t=0}^{\infty} \gamma^t R(s_t, a_t, s_{t+1}, g) \right] \tag{1}$$

where $\gamma \in [0, 1]$ is a discount factor assigning larger weights to more immediate rewards. We consider the special case where the reward function is sparse and given by an indicator function that only depends on the next state and the goal:

$$R(s_t, a_t, s_{t+1}, g) = \mathbb{1}(s_{t+1}, g) = \begin{cases} 1, & \text{if } s_{t+1} \text{ achieves } g, \\ 0, & \text{otherwise} \end{cases} \tag{2}$$

i.e. we have some criteria that tells us whether any given state $s$ achieves any given goal $g$, and only provide a reward when this is satisfied.

### 3.2 Hindsight Experience Replay (HER)

In complex environments it is extremely unlikely that the specified goal $g$ will ever be achieved by chance. As such, standard RL algorithms struggle in sparse-reward, multi-goal environments because they receive very little learning signal from which they can improve their policy. The key insight of HER is that trajectories that don't achieve the specified goal still contain useful information about how to achieve *other* goals — namely those that are observed later on during the same trajectory. By using an off-policy RL algorithm such as DQN [34] or DDPG [35] it is possible to re-label samples

that were collected by the policy whilst attempting to achieve a goal $g$ with an alternative goal $g'$, and subsequently re-compute the reward. For example, if $(s_t, a_t, r_t, s_{t+1}, g)$ is sampled from a replay buffer of past experience, $g$ can be replaced with another goal $g'$ that occurs later in the trajectory, and then a reward for this new goal can be recomputed: $r'_t = R(s_t, a_t, s_{t+1}, g')$. This new transition can still be used in training an off-policy RL algorithm since the original goal only influences the agent's action, but not the dynamics of the environment. By re-labelling transitions this way HER can significantly speed up the learning of a goal-conditioned policy since it increases the frequency with which the transitions seen in training actually lead to the specified goals being achieved.

## 4 Methods

The key insight of our method is that the same principle underlying HER — i.e. that any observed trajectory contains useful information about how to achieve the goals observed during that trajectory — has the potential to be used more efficiently as part of a model-based algorithm. In particular, instead of re-labelling transitions and re-computing rewards, we propose to make more complete use of the information contained within the observed transitions by training a generative model that can generate *plausible transitions* leading from the current state towards a desired goal. That is, we use experience gathered by the agent to train a goal-conditioned model that can generate future trajectories (states and actions) that move the agent towards any goal that we specify. These imagined trajectories do not necessarily need to be optimal in the sense of moving directly towards the goal, since the second key component of our method involves feeding these proposed trajectories into a planning algorithm that decides which action to take in order to achieve the goal in as few steps as possible.

Whilst in principle a number of generative models could be used for this purpose, in this work we choose to use GANs [14], since they can easily deal with high-dimensional inputs and do not explicitly impose any restrictions on the form of the distribution produced by the generator. Specifically, we choose to use WGANs (Wasserstein GANs) [36] with spectral normalisation [37], as recent work has shown that these can be trained in a stable manner even when the underlying training data is non-stationary [24].

### 4.1 Training the GAN(s)

The aim of the first major component of our method is to train a generative model that can take in the current state $s_t$ along with a desired goal $g$ and produce an imagined action $a_t$ and next state $s_{t+1}$ that moves the agent towards achieving $g$. We approach this by training an ensemble of $N$ conditional-GANs, each consisting of a generator $G_{\phi_i}$ and a discriminator $D_{\theta_i}$ where $\{\theta_i\}_{i=1}^N$, $\{\phi_i\}_{i=1}^N$ are the parameters of the neural networks that represent these functions. The generators take in the current state $s_t$, a noise vector $z$ and the target goal $g$ in order to produce an imagined action $a_t$ and next state $s_{t+1}$. The discriminators take in $s_t$, $a_t$, $s_{t+1}$ and $g$ and aim to distinguish whether or not this is a transition from a real trajectory that eventually reaches goal $g$ or an example created by the generator.

We also consider a variation where concurrently we train an ensemble of $N_m$ deterministic one-step predictive models of the environment. The aim of these predictive models is to take a state-action pair $(s_t, a_t)$ and predict the difference between the next state and the current state, $s_{t+1} - s_t$, as in [15]. We denote these models as $f_{\beta_j}$, where $\{\beta_j\}_{j=1}^{N_m}$ represent the parameters neural networks representing these functions. These predictive models can be used to provide an L2 regularisation term in the generator loss that encourages the generated actions and next states to be consistent with the predictions of the one-step models — although this is not necessary to make the method work (we study the effect of using predictive models this way in Section 5). The whole setup is shown schematically in Figure 1.

The loss for the $i^{th}$ generator is as follows:

$$\mathcal{L}_{\text{generator}}^{(i)} = \mathbb{E}_{\substack{z \sim p(z) \\ s_t, g \sim \mathcal{R} \\ s_{t+1}, a_t \sim G_{\phi_i}(z, s_t, g)}} \left[ D_{\theta_i}(s_t, g, s_{t+1}, a_t) + \lambda \frac{1}{N_m} \sum_{j=1}^{N_m} ((s_{t+1} - s_t) - f_{\beta_j}(s_t, a_t))^2 \right]$$

(3)

where $\mathcal{R}$ is a replay buffer of real experienced trajectories, $z \sim p(z)$ is a noise vector where each component is sampled independently from the standard normal $\mathcal{N}(0, 1)$ and $\lambda$ is a parameter that weights how strongly we penalise deviations in the generated action/next state from the average predictions made by one-step models. The loss for the $i^{th}$ discriminator is:

$$\mathcal{L}_{\text{discriminator}}^{(i)} = \mathbb{E}_{\substack{s_t, a_t, \\ s_{t+1}, g}} \sim \mathcal{R} \left[ D_{\theta_i}(s_t, g, s_{t+1}, a_t) \right] - \mathbb{E}_{\substack{z \sim p(z) \\ s_t, g \sim \mathcal{R} \\ s_{t+1}, a_t \sim G_{\phi_i}(z, s_t, g)}} \left[ D_{\theta_i}(s_t, g, s_{t+1}, a_t) \right] \quad (4)$$

The replay buffer $\mathcal{R}$ is populated initially by random trajectories, however we find it helpful to filter (i.e. not store) trajectories where the final achieved goal is identical to the initial achieved goal, since these provide nothing useful for the GANs to learn from. After some initial training further trajectories generated by the planner (described in the next section) are also added to $\mathcal{R}$ whilst training continues, allowing for continuous, open-ended improvement. Note that this makes the data distribution we are trying to emulate non-stationary as new self-collected data is constantly being added. The sampled goals from the replay buffer are always taken as goals achieved at a randomly chosen time step that occurs later within the same trajectory.

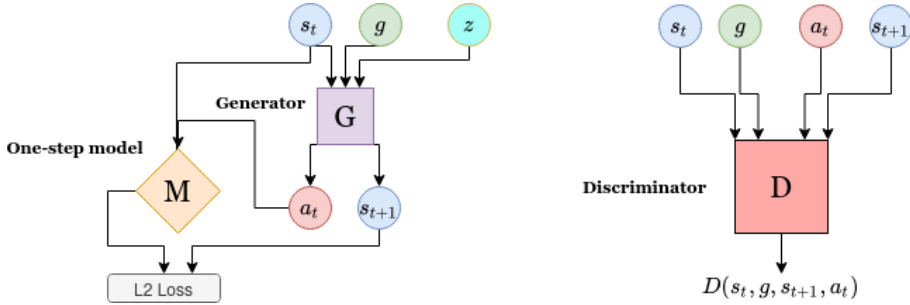

Figure 1: Structure of the generative model used in PlanGAN.

The basic building block is a generator that takes a state, goal and noise vector and produces an action and next state. However, during training we actually generate trajectories consisting of $\tau$ time steps. That is, we take the generated state from the previous step and use this as input to the generator to produce a new action/next state pair, and repeat. The generator is then trained by backpropagating through these unrolled trajectories. In more detail, we sample batches of real trajectories made up of $\tau$ transitions from the buffer: $(s_0, a_0, g_0, s_1, a_1, g_1, \ldots, s_{\tau-1}, a_{\tau-1}, g_{\tau-1}, s_\tau)$, where each goal $g_i$ is an achieved goal at a later time along that same trajectory (we found that choosing a different goal at each time step worked better than just a single goal for the whole trajectory). We then use the generator to generate a trajectory $(\hat{s}_0 = s_0, \hat{a}_0, g_0, \hat{s}_1, \hat{a}_1, g_1, \ldots, \hat{s}_{\tau-1}, \hat{a}_{\tau-1}, g_{\tau-1}, \hat{s}_\tau)$, where $\hat{s}_t, \hat{a}_{t-1} = G_\phi(z_t, \hat{s}_{t-1}, g_{t-1})$. Batches of these real and imagined trajectories are then used to calculate the expectations in the losses shown in Equations 3 and 4. Training end-to-end on sequences of transitions imposes more constraints on the generator, requiring full trajectories to be difficult for the discriminator to distinguish rather than just individual transitions, and is crucial for good performance.

Each GAN and one-step model in the ensemble has a different random initialisation and is trained on different batches of data sampled from the same replay buffer. As discussed in the context of using an ensemble of one-step models for model-based RL [17], this is enough to give the models significant diversity. We study the benefits of using an ensemble over a single GAN in the Section 5.

## 4.2 Planning to achieve a goal

Once we have an ensemble of GANs that has been trained on some amount of real data, we use these to plan the actions to take in the environment to achieve a given goal, $g$. Our planner's basic structure shares similarities with a number of other model-predictive control based approaches [15, 16, 17, 38] — make use of a model to generate a number of imaginary future trajectories, score them, use these scores to choose the next action, and repeat this whole procedure at the next step. The novelty in our approach is in the fact that our trajectories are generated using GANs, the way that we score the trajectories and how we make use of an ensemble of models.

To plan the next action to take from the current state $s_t$ towards a desired goal $g$, we first sample a set of $Y$ initial actions and next states, $\{a_t^y, s_{t+1}^y\}_{y=1}^Y$. For each $y$, $a_t^y$ and $s_{t+1}^y$ are generated from a random generator in the ensemble, conditioned on $s_t, g$, i.e. $a_t^y, s_{t+1}^y = G_{\phi_i}(s_t, g, z)$, where $i \sim \text{Uniform}\{1, \ldots, N\}$. Our aim is then to give each of these initially proposed actions a score which captures how effective they are in terms of moving towards the final goal $g$. A good score here should reflect the fact that we want the next action to be moving us towards $g$ as quickly as possible whilst also ensuring that the goal can be retained at later time steps. For example, we would not want to score too highly an action that moved an object close to the desired goal with very high velocity such that it would overshoot and not remain there at later time steps.

To obtain such a score we duplicate each of the $Y$ initial seed actions and next states $C$ times. Each next state $\{s_{t+1}^{y,k}\}_{y=1,k=1}^{Y}{}_{}^{C}$ is then used as the starting point for a trajectory of length $T$. These hypothetical trajectories are all generated using a different randomly chosen GAN at each time-step, so for example $s_{t+w}^{y,c}$ is generated from a random generator in the ensemble conditioned on $(s_{t+w-1}^{y,c}, g)$.

Once we have generated these trajectories, we give each of them a score based on the *fraction of time they spend achieving the goal*. This means that trajectories that reach the goal quickly are scored highly, but only if they are able to remain there. Trajectories that do not reach the goal within $T$ steps are given a score of zero. We can then score each of the initial seed actions $\{a_t^y\}_{y=1}^Y$ based on the *average score of all the imagined trajectories that started with that action*. These scores are normalised and denoted as $n_y$, and we define weights $w_y = e^{\alpha n_y}$, where $\alpha > 0$ is a hyperparameter. The final action returned by the planner is either the action with the maximum score or an exponentially weighted average of the initially proposed actions, $a_t = \frac{\sum_{y=1}^Y w_y a_y}{\sum_{y'=1}^Y w_{y'}}$. The rationale for using a different random generator at each step of every hypothetical trajectory is that we will be giving higher scores to initial actions that all of the GANs agree can spend a lot of time achieving the goal. This improves the robustness of the predictions and protects against errors in terms of unrealistic imagined future trajectories generated by any single GAN.

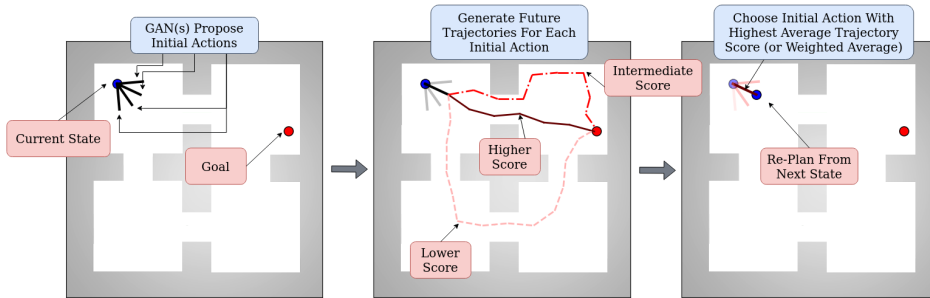

Figure 2: Illustrative example of how the planning algorithm works.

## 5 Experiments

We perform experiments in four continuous environments built in MuJoCo [39] — Four Rooms Navigation, Three-link Reacher, Fetch Push and Fetch Pick And Place (see Figure 3). Full details about these environments, along with the hyper parameters used for the experiments, can be found in the Appendix. We evaluate the performance in terms of the percentage of goals that the agent is able to reach vs. the number of time steps it has interacted with the real environment[1].

### 5.1 Comparisons

We have compared the performance of our algorithm against a number of leading model-free methods designed for multi-goal, sparse reward environments (Figure 4). The most natural baseline to compare

**Algorithm 1:** PlanGAN

**initialise:** generators $\{G_{\phi_m}\}_{m=1}^M$, discriminators $\{D_{\theta_m}\}_{m=1}^M$, one-step models $\{f_{\beta_k}\}_{k=1}^K$, replay buffer $\mathcal{R}$, environment Env

**begin**

    **for** $j = 1 : J$ **do**
        Append random trajectory $(s_0, a_0, g_0, \ldots, a_{T-1}, s_T, g_T)$ to $\mathcal{R}$

    **for** $y = 1 : Y$ **do**
        train()

    **for** $e = 1 : E$ **do**
        Sample goal $g$ from environment
        $(s_0, a_0, g_0, \ldots, s_T, g_T) =$ planner($g$)
        Append $(s_0, a_0, g_0, \ldots, s_T, g_T)$ to $\mathcal{R}$
        **for** $p = 1 : P$ **do**
            train()

**procedure** train()

    **for** $m = 1 : M$ **do**
        Sample batch of $B_g$ trajectories from $\mathcal{R}$:
        $(s_0, a_0, \hat{g}_0, \ldots, \hat{g}_{\tau-1}, s_\tau)_{b=1}^{B_g}$
        Use $G_{\phi_m}$ to generate a batch of $B_g$ imagined trajectories, starting from the real $s_0$ values and conditioning on the same goals $\hat{g}_0, \ldots, \hat{g}_{\tau-1}$ as in the real trajectories
        Train $G_{\phi_m}, D_{\theta_m}$ with equations 3 and 4

    **for** $k = 1 : K$ **do**
        Sample batch of $B_m$ transitions from $\mathcal{R}$:
        $(s_t, a_t, s_{t+1})$
        Train $f_{\beta_k}$ to minimise:
        $\mathbb{E}\left[||f_{\beta_j}(s_t, a_t) - (s_{t+1} - s_t)||_2^2\right]$

**procedure** planner($g$)

    $s_0, g_0 \longleftarrow$ Env.reset(); Trajectory $= (s_0, g_0)$
    **for** $t = 0 : T - 1$ **do**
        InitAcs $= \{\}$; Scores $= \{\}$
        **for** $y = 1 : Y$ **do**
            $i \sim$ Uniform$(1, \ldots, M)$
            $z = [z_k]_{k=1}^d, z_k \sim \mathcal{N}(0,1)$
            $\hat{s}_{t+1}^y, \hat{a}_t^y = G_{\phi_i}(z, s_t, g)$
            InitAcs.append($\hat{a}_t^y$)
            ImaginedTrajs $= \{\}$
            **for** $c = 1 : C$ **do**
                $s_{t+1}^{y,c} = s_{t+1}^y$
                **for** $t' = t + 1 : t + T$ **do**
                    $i \sim$ Uniform$(1, \ldots, M)$
                    $z = [z_k]_{k=1}^d, z_k \sim \mathcal{N}(0,1)$
                    $\hat{s}_{t'+1}^{y,c}, \hat{a}_{t'}^{y,d} = G_{\phi_i}(z, \hat{s}_{t'}^{y,c}, g)$
                ImaginedTrajs.append($\hat{s}_{t+1}^{y,c}, \ldots, \hat{s}_{t+T}^{y,c}$)
            score$[y] = \frac{1}{T+1} \sum_{t'=t}^{t+T} \mathbb{1}(\hat{s}_{t'}^{y,c}, g)$
            scores.append(score$[y]$)
        scores = Normalise(scores)
        $a_t = \frac{\sum_{y=1}^Y e^{\alpha \text{ scores}[y]} \hat{a}_t^y}{\sum_{y'=1}^Y e^{\alpha \text{ scores}[y']}}$
        $s_{t+1}, g_{t+1} =$ Env.step($a_t$)
        Trajectory.append($a_t, s_{t+1}, g_{t+1}$)
    **return** Trajectory

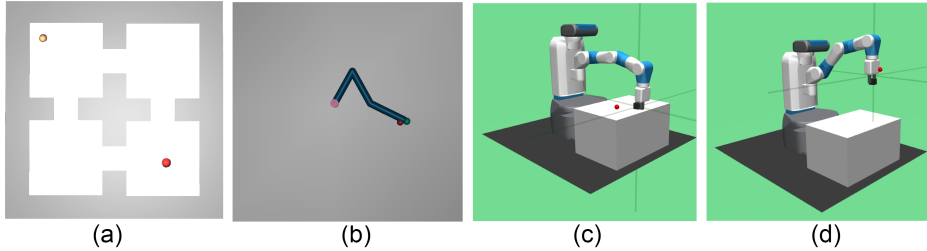

Figure 3: Environments that we evaluate PlanGAN on. (a) Four rooms navigation. (b) Reacher (Three Links). (c) Fetch Push. (d) Fetch Pick And Place.

with is HER (using DDPG as the core RL algorithm [8]), as this is based on a similar underlying principle to PlanGAN. We also include DDPG without HER to demonstrate how standard model-free RL methods struggle with these tasks. For both of these we use the implementations found in OpenAI Baselines [40]. We also include comparisons with two recently proposed modifications to HER, "Curriculum-Guided HER" [30] (CHER) and "Soft HER" [32][2] (SHER). We also include a model-based baseline (PETS)[41], which also makes use of ensembles of models but which is not designed specifically with multi-goal, sparse reward tasks in mind. Note that it is computationally prohibitive to run this method for as long as the model-free methods, however we run it for at least as many steps as we run PlanGAN. Finally, for the Fetch Push and Fetch Pick And Place environments,

we include comparisons with a recent method "Simulated Locomotion Demonstrations" (SLD) [42], which requires an object to be defined. SLD uses the fact that with a simulator objects can move by themselves, so a separate object policy can be learned where the object moves itself to the desired goal. SLD leverages this object policy to guide the learning of the full robot policy. This gives it a significant advantage over PlanGAN as it makes use of separately learned self-generated demonstrations to guide the training, however we see that PlanGAN still achieves significantly better data efficiency. All plots are based on running each experiment using 5 different random seeds, with the solid line representing the median performance and the shaded area representing one standard deviation around the mean. We also include a line showing the average asymptotic performance of HER (as this is the most directly comparable method). Note that the environment interactions recorded on the training curves for PlanGAN *do* include both the initial random trajectories as well as any trajectories that are not stored in the buffer (when the final goal is identical to the initial goal).

In all of the tasks considered we find that PlanGAN is significantly more sample efficient than any of the other methods model-free methods, requiring between 4-8 times less data to reach the same performance as HER. This is comparable to the sample efficiency gains reported in [15] for a model-based approach to dense reward tasks over leading model-free methods. It also substantially outperforms the model-based baseline (PETS) which is not designed for sparse reward, multi-goal environments.

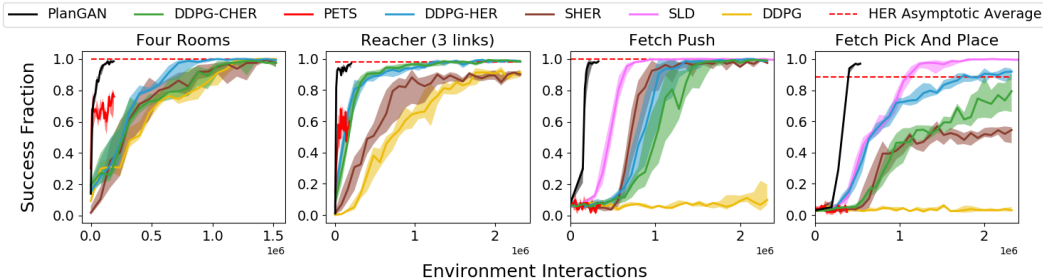

Figure 4: PlanGAN performance compared to existing methods.

## 5.2 Ablation studies

In this section we study how various decisions we have made affect PlanGAN's performance by performing ablation studies on the two more complicated environments considered (Fetch Push and Fetch Pick And Place). Firstly, we study whether the planner is a crucial component of our set-up. The first panel in Figure 1 in the Appendix shows a comparison of the full PlanGAN with a couple of variations that more directly use the actions proposed by the GANs. Both of these lead to significantly lower success rates, suggesting that the planner we use is crucial.

We then consider how the number of GANs in the ensemble effects PlanGAN's performance. The second panel in Figure 1 (Appendix) shows results for ensembles made up of 1, 3 and 5 GANs respectively. Whilst less significant than the inclusion of the planner, we find that using only a single GAN leads to slower and significantly less stable training. We also see that the larger ensemble (5 GANs) outperforms the smaller ensemble (3 GANs), but the difference in performance is relatively small. Finally, we consider running the algorithm with $\lambda = 0$, i.e. without any regularisation from the one-step predictive model. We see that the one-step model regularisation provides only a very minor improvement, suggesting that it is not a crucial component of PlanGAN.

## 6 Conclusions

We proposed PlanGAN, a model-based method for solving multi-goal environments with sparse rewards. We showed how to train a generative model in order to generate plausible future trajectories that lead from a given state towards a desired goal, and how these can be used within a planning algorithm to achieve these goals efficiently. We demonstrated that this approach leads to a substantial increase in sample efficiency when compared to leading model-free RL methods that can cope with sparse rewards and multiple goals.

In the future we would like to extend this work so that it can be applied to more complex environments. One of the main limitations with the current approach is the planner. When the number of time steps required to complete a task becomes large the planner becomes computationally expensive, since at each step we have to simulate a large number of future steps out until completion. We also need these trajectories to be at least reasonably accurate over a large number of time steps, as imagined future trajectories that do not reach the desired goal are given a score of zero. If no imagined trajectories reach the goal then the planner is unable to meaningfully choose an action. Future work which may more efficiently deal with longer horizon tasks could involve combining the GAN training with a model-free goal-conditioned value function (creating a hybrid method, similar to STEVE [19] and Dreamer [7]) which could learn to give a value to the actions proposed by the GANs, removing the need for a planner entirely.

## Statement of Broader Impact

Since our work involves foundational research in the field of model-based reinforcement learning it is unlikely to have any large, immediate impacts on society. Nevertheless, in the longer term the impact of reinforcement learning agents capable of learning to autonomously make decisions could be huge. In principle one could discuss a huge range of potential impacts over different time frames, but we choose to focus on some potential medium-term impacts of robots that can learn autonomously from sparse rewards.

Robots are pervasive in the modern world and are used in a wide range of manufacturing industries for carrying out tedious, repetitive work. Introducing robots that are capable of autonomously learning a set of skills from easy to specify reward functions has the potential to vastly increase the scope of possible tasks that they can be used for. In particular, it removes the requirement for their behaviours to be carefully engineered in a manual fashion for every possible scenario they might encounter. This has the potential to allow for many tasks that currently can only be carried out by human workers to become fully or partially automated.

Whilst this could provide a huge economic boost to some manufacturing companies, it is important that this benefit is weighed against the potential negative impacts (both social and economic) that losing these manufacturing jobs could have — particularly if large scale changes were to occur in a short period of time. We feel that this is an important question for both economists/ policy advisors as well as researchers working in the field to think about.

## Acknowledgements/ Funding Disclosure

This work was partially funded by Catapult (High Value Manufacturing) within Warwick Manufacturing Group.

## Footnotes

[1]Videos of results are available here: https://sites.google.com/view/plangan/home

[2]using the official implementations found here and here respectively

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
