[Supplementary Material]

## A  Experimental Details

### A.1  Environments

**Four Rooms Navigation:** A point mass is placed within a miniature four-room maze. Both the state space (four-dimensional - position and velocity of the mass) and the action space (two-dimensional - acceleration in x/y direction) are continuous. Goals are two dimensional (target position) and are sampled uniformly at random.

**Reacher (Three Links):** Three links are connected together with hinges. The agent must apply torques to these hinges to move the end-point to the specified goal. The state space is 11-dimensional, action-space 3-dimensional and the goal-space 2-dimensional.

**Fetch Push:** A robotic arm (with its gripper forced shut) interacts with a cube. The aim is to push the cube to a desired goal position. The state space is 21-dimensional (robot positions/velocities, object position/velocity) and the action-space is 4-dimensional. The goal space is the position of the cube (3-dimensional), and goals are sampled uniformly over a region on the table.

**Fetch Pick And Place:** The robotic arm is the same as Fetch Push, but now the gripper can also be controlled and opened. The aim is to pick up the cube and move it to a desired goal. The state space is 25-dimensional and the action-space is 4-dimensional. The goal is the position of the cube and can also be in the air (above the table), such that the gripper must be used to pick the cube up.

### A.2  Hyperparameters

The hyperparameters used were largely the same for all of the experiments reported. Here we give a list and description of them, as well as their default values.

| Symbol in paper | Description | Default values |
|---|---|---|
| $M$ | Number of GANs in ensemble | 3 |
| $K$ | Number of one-step predictive models | 3 |
| $J$ | Number of initial random trajectories stored before we train | 250 |
| $Y$ | Number of initial training steps | 100000 |
| $T$ | Trajectory length of gathered rollouts (and planning horizon) | 50 |
| $E$ | Number of additional trajectories gathered (after initial random) | 1500 (Four Rooms, Reacher), 2000 (FP), 3000 (FPAP) |
| $P$ | Number of training steps per new trajectory gathered | 250 |
| $Q$ | Number of initial actions proposed during planning | 25 |
| $C$ | Number of copies of each initial action during planning | 100 |
| $\tau$ | Length of trajectories that we sample/generate during training | 5 |
| $\lambda$ | Regularisation strength of the one-step predictive model in the generator loss | 30.0 |
| $B_g$ | Batch size for training GANs | 128 |
| $B_m$ | Batch size for training the one-step models | 256 |
| $\alpha$ | Parameter for weighting trajectory scores | 5 (Four Rooms, FPAP) N/A (max score) (FP, Reacher) |
| - | Generator optimiser | ADAM(lr=0.0001, $\beta = (0.5, 0.999)$) |
| - | Discriminator optimiser | ADAM(lr=0.0001, $\beta = (0.5, 0.999)$) |
| - | One-step model optimiser | ADAM(lr=0.001, $\beta = (0.9, 0.999)$) |
| - | Generator parameters L2 regularisation | 0.0001 |
| - | Discriminator parameters L2 regularisation | 0.0001 |

429  Note that for Fetch Push and Fetch Pick And Place, the number of initial trajectories $J$ does not correspond to
430  $J \times T$ environment interactions, as many random trajectories do not move the object and hence do not change
431  the achieved goal, so are not stored in the buffer. Here $J$ refers to the number of initial trajectories that are
432  actually stored in the replay buffer (although the discarded trajectories are still counted when reporting the
433  number of environment interactions).

### A.3   Network details

435  All of the generator, discriminator and one-step predictive models consist of fully connected neural networks
436  with two hidden layers of size 512. The hidden layers in the generator have BatchNorm applied to them. Apart
437  from the output layers all activations are ReLU.

## B   Ablation Studies

439  We performed ablation studies on the two more challenging environments considered (Fetch Push and Fetch
440  Pick And Place). Firstly we aimed to address how necessary the use of the planner was. To do this we carried
441  out experiments where we just trained the GANs and then used their proposed actions directly to generate new
442  trajectories. We considered two variations — firstly simply choosing the first action proposed by a random GAN
443  in the ensemble (NoPlanner in the first panel of Figure 5), and the second where we take the average over a large
444  number of actions proposed by the ensemble of GANs without scoring them (NoPlannerAvg in the first panel).

445  The next question we addressed was how the number of GANs in the ensemble impacts performance. We run
446  experiments for a small ensemble (3 GANs, and the "standard setting"), a larger ensemble (5 GANs) and no
447  ensemble (1 GAN). The results are shown in the second panel of Figure 5. We see that no ensemble leads to
448  slower and less stable training, particularly for Fetch Pick And Place. We see that a larger ensemble does lead to
449  an improvement over the smaller ensemble, although the difference is relatively small.

450  Finally we consider running the experiment without any regularisation from the one-step predictive model
451  (NoOSMReg), i.e. $\lambda = 0$. We see that this only leads to a very minor decrease in performance, suggesting that
452  the inclusion of a one-step predictive model is not really a crucial component of PlanGAN.

Figure 5: Ablation results

453  Note that other than the described changes the parameters used for each experiment were as described in
454  Appendix A.