[Reviews · NeurIPS 2020]

Review 1

Summary and Contributions: The paper presents a model-based goal-conditioned reinforcement learning method. In contrast to the commonly used model-free method, HER, the proposed method uses a goal-conditioned predictive model of dynamics p(s',a|s,g) and a shooting-based planning method. The model is trained on off-policy data from the replay buffer. It is shown that this method reaches the asymptotic performance of the model-free HER, while being several times more data-efficient. --- Decision --- The paper presents an approach for model-based goal-conditioned reinforcement learning which works well in practice and is more data-efficient that the model-free state-of-the-art. While there are unresolved questions about theoretical interpretation and comparison to prior work, I believe the paper is likely to be impactful. I recommend acceptance. --- Update --- The rebuttal does not require a response. I urge the authors to perform the crucial comparison to a model-based method such as PETS.

Strengths: The paper proposes an interesting model-based method for an important problem of goal-conditioned reinforcement learning. The method empirically performs well, and is more data-efficient than existing model-free methods, suggesting possible wide adoption.

Weaknesses: There are two main weaknesses of the paper: lacking theoretical analysis and missing model-based baselines. First, the paper contains no theoretical analysis. It is thus unknown whether the method is expected to converge to the optimal solution. Intuitively, since the learned distribution of trajectories is fit to the replay buffer, the samples will be biased in a certain way. As the planning averages the samples, it is not guaranteed to converge to the optimal plan. Furthermore, it is unclear what objective the model is optimizing, as well as what objective the planning is optimizing. This will prevent future researchers from improving the method as it's unclear what the basic principles of designing a model-based goal-conditioned method are. The paper contains an extensive comparison to prior model-free work such as HER or DDPG. However, there is no comparison to any model-based method. Some appropriate comparisons would be Chua'18 or Nasiriany'20 Chua'18, Deep Reinforcement Learning in a Handful of Trials using Probabilistic Dynamics Models.

Correctness: Yes

Clarity: Some internal references are broken, see l273, l277. These are seemingly intended to refer to Fig. 5 in the appendix.

Relation to Prior Work: The paper misses a large prior literature on planning with a specified goal, e.g. Kurutach'18, Pathak'18, Liu'20, Nair'20. Also quite confusingly, the paper compares to a method called "Simulated Locomotion Demonstrations", but does not contain a reference to it. Kurutach'18, Learning plannable representations with causal infogan Pathak'18, Zero-Shot Visual Imitation Liu'20, Hallucinative Topological Memory for Zero-Shot Visual Planning Nair'20, HIERARCHICAL FORESIGHT: SELF-SUPERVISED LEARNING OF LONG-HORIZON TASKS VIA VISUAL SUBGOAL GENERATION

Reproducibility: Yes

Additional Feedback:


Review 2

Summary and Contributions: The paper introduces an methods that combines GAN model and planning to dramatically improve sample efficiency on multiple goal environments. Section 1 presents the motivations and the proposed approach. Section 2 gives an overview of related work. Sections 3 gives more details on the target task for this method (multiple goal RL environments), and gives a bit more detail on a successful method in this field (HER) which the proposed approach builds upon. Section 4 describe the proposed method and how exactly the GANs are trained in the and how they are used inside a planner. Section 5 presents experiments which convincingly show that the proposed approach is much more data efficient that previous methods.

Strengths: The paper is straightforward, easy to read. The motivations are clear, the proposed method is precisely described and the experiments are convincingly showing the greater data efficiency of the proposed approach and plausible.

Weaknesses: While the proposed approach seem to be much more data efficient, it might not be more compute efficient. For example PlanGAN curves in figure 4 stop much earlier in environment interactions than other methods, suggesting that it was prohibitive to run for as many steps as previous methods. Many of the choices that were made seem somewhat arbitrary and are not backed by empirical evidence or convincing intuition. For example in the planner, the GAN used to generate imaginary rollouts is swapped at every time step. Also during GAN training, the goal is random amongst goals that have been subsequently achieved, many other choices could have been made. The planner consisting of Q suggested seed action followed by C imaginary rollouts for each action is bit naive and limited because the rollout have to be as long as the time horizon for reaching the goal. Would have been interesting to use a more sophisticated planner that can act on smaller horizon with value such as MCTS/MuZero style. Finally, combining GAN model and planning is in my opinion a very interesting idea but I am not sure why they author chose to only apply it to multiple goal setup. I think applying it to the less specialised environment would have been much more interesting and impactful. For all these reasons (a bit too narrow focus on multi-goal, lack of grounding to prior work results for the claim about data efficiency, naive planner, some arbitrary choices), I think the paper is not as impactful as it could be have been, although the idea being explored is quite interesting, hence my reserved rating.

Correctness: Most of the claims and equations were correct except maybe the way actions are sampled in the planner which does not make sense to me (see additional feedback). The experiment look convincing enough but the baselines have been rerun by the author without any kind of anchoring in previously published numbers, excepts for HER asymptotic performance. So the claim about better sample efficiency is not really grounded in any external numbers.

Clarity: The paper is very straightforward and easy to read. There are a few inaccuracies here and there, see additional feedback section for more details.

Relation to Prior Work: Yes prior work is clearly described, and it is clear how this work differs from previous contributions.

Reproducibility: Yes

Additional Feedback: l195: "we take the generated ... and repeat. The generator is then trained on these end-to-end" : maybe terms like "unroll" and "back-propagate through time" might be clearer and more precise. l199. I guess different strategies are possible here. Maybe having a single future achieved goal in the unroll would also work? Or piece-wise constant goals? Or always have the next achieved goal? Have you tried any of these? Any reasons not to? l217 Q might not be a great notation in the context of deep RL as it is usually reserved to action value function? l225: maybe this can be more concise clearer like "Q seed actions and C imaginary rollouts for each of these seeds". l227: If I understand you are potentially swapping the GAN model at every time step in each rollout? This might be a bit out-of-distribution since each GAN model has been trained only on their own rollouts, not interleaved rollouts? Maybe either training the ensemble like that or using a fixed model in each rollout would do better? l235 The weighted averaged action does not really make sense to me as a selected action (not an integer, depends on the ordering etc...). Also why does the sum on the numerator goes from 1 to R and 1 to Q on the denominator? Maybe you mean the selected action is a sample from the softmax of the scores of each seed action? That would make more sense. l237-239: as explained above I am not sure I fully agree with the intuition here. Algorithm 1: again the weighted average in planner(g) still doesn't make sense to me (maybe replace with sample of softmax of normalised scores?). l273 & l277: broken reference "Figure ??". Not great to spend significant main text describing figures in the appendix. If you think these are important (I would agree), maybe move those in the main paper? Moving the pseudo code (which does not bring much in my opinion and takes a lot of space) to the appendix might help. l291 yes I agree, the planner seems like a big limitation, would be interesting to see how such method does inside a more sophisticated planner which doesn't require to unroll to goal completion to get some signal, like MCTS/MuZero for example.


Review 3

Summary and Contributions: The authors are interested in solving goal-directed tasks with sparse rewards. The authors propose to train a goal-conditioned dynamics model that can easily next-state predictions. Specifically, they train an ensemble of goal-conditioned GANs to generate the next state and action conditioned on a goal and current state. These one-step GAN models use their own predictions to generate trajectories. The GAN is trained to discriminate between generated and real trajectories, where the goal is relabelled with some achieved goal. To choose actions, the authors then sample a number of actions and trajectories given the current state, and choose the actions that lead to trajectories that have the highest score. (The score is the fraction of states that achieve some goal). The authors demonstrate across a variety of robot tasks that the method performs significantly better than model-free methods such as HER, curriculum HER, and soft HER.

Strengths: The empirical results of the paper are very strong and consistently outperform model-free method on sparse reward tasks. These impressive results are likely to be of interest to researchers studying goal-conditioned reinforcement learning and model-based learning.

Weaknesses: While the empirical results are strong, the paper would be strengthened with a comparison to a model-based method that is not goal-conditioned. In fact, given that the main claim at the end of the introduction is that their method is “explicitly designed for multi-goal, sparse reward” and therefore “[leads] to a significant improvement in sample efficiency,” this seems like a very important comparison to run. The paper would be also strengthened by testing the method on some more difficult domains, such as the “hand” environments that have higher intrinsic dimension, and where model-free methods generally excel.

Correctness: One detail that seems important is whether or not the training curves for the PlanGAN method take into account the random trajectories at the beginning of training. I also suggest a few additional ablations (see Additional feedback).

Clarity: Overall, the paper is written well. Some minor comments: What is M in Figure 1? Section 5.2 has broken references. Given that the OSM regularization had such little impact (Figure 5 in the Appendix), I’m surprised that the author did not remove that from the paper. Is there a use for it that’s not discussed in the paper?

Relation to Prior Work: While the authors have a nice summary of related works, the related works section would be greatly improved if the authors discussed specifically how these prior methods relate to this paper. The authors should also discuss the relationship to [1] and ideally compare against it. The two methods seem rather similar, with one important difference being that this method proposes to stochastically generate next states and actions, rather than predict only states. [1] Goal-Aware Prediction: Learning to Model What Matters. Nair, Suraj et al. ICML 2020.

Reproducibility: Yes

Additional Feedback: When generating rollouts, do the authors “mix” the generators in ensemble? In particular, given an initial state s, action a, and goal g, I use generator 1 to sample a new state s_2 and new action a_2. To generate state s_3 and action a_3, do I use generator 1 again, or do I randomly sample a new generator? From “These hypothetical trajectories are all generated using a different randomly chosen GAN at each time-step, so for example q,c s q,c t+w is generated from a random generator in the ensemble conditioned on (s t+w−1 , g).“ it sounds like there is mixing going on. If so, this seems like an interesting, but rather non-conventional technique. How important was this design decision? It would also be good to see an ablation that also shows how important the trajectory-based loss is, given that the author say that it is “crucial for good performance.” How sensitive is the method to the parameter lambda? -- I read the author rebuttal. Thank you for clarifying. I am keeping my score.


Review 4

Summary and Contributions: This paper proposes to use GAN to solve goal-conditioned Reinforcement Learning tasks. Conditioning on the goal, their GAN can predict the current action and the next state based on the current state. They train an ensemble of several GANs on the relabeled trajectories. They then do model-based planning with the stochastic generative models and achieve 4-8 times more sample efficiency. --- Update --- The author rebuttal addressed my issue partially. I appreciate the demonstration that shows HER's performance degrades w.r.t. num_updates. I agree that it's unnecessary to compare with those variants I mentioned. Still, the paper could be more solid if the authors can study more thoroughly and make fairer comparisons, which can help us understand each component's benefit better. I maintain my score.

Strengths: This work integrates the recent achievement in the deep generative model into the goal-conditioned reinforcement learning. They train the goal-condition model in a hindsight way to solve the sparse reward problem, which combines the advantage of planning method and HER and achieves high sample efficiency. The improvement is very significant and will inspire future works.

Weaknesses: As mentioned in conclusion, the experimental domain is not very challenging. Those domains have a simple forward model, and the planning horizon is typically very short. The proposed model-based approaches may take advantage of that.

Correctness: Yes.

Clarity: The paper is clear.

Relation to Prior Work: Yes

Reproducibility: Yes

Additional Feedback: I have several minor issues: 1. The proposed methods require a more significant number of gradient descent to train the ensemble model. Am I right? What if we do more updates per epoch for HER or other baselines? Sometimes increasing the num_update_per_epoch may increase the data-efficiency in the goal-conditioned RL. 2. What is the performance of HER if we learn an ensemble of the policies? 3. I wonder if we can combine HER with a forward model and do a model-based search. This should be a very straightforward baseline to compare. 4. The figure indices on line 273 and 277 are missing. I would raise my score if the author can address my issues.

[Author Response · NeurIPS 2020]

Firstly we would like to thank all of the reviewers for their insightful and thorough feedback. We really appreciate the time taken to go through the paper and will try our best here to respond to as many of the questions raised as possible.

To start with, three of the reviewers noted that we had not included any comparisons with other model-based approaches. Our initial reasoning for not including such a comparison was that because standard model-based methods generally do not outperform model-free methods in terms of their final performance, it seems implausible that a model-based approach designed for standard environments (i.e. dense reward, not goal-based) would perform better than, e.g., DDPG, which we do include as a comparison and which performs badly on the more challenging environments. Having thought about this more, however, it does seems natural that such a baseline should still be included to verify this intuition, and we will look to add such a baseline in the next draft. We also thank **reviewer 1** for drawing our attention to a number of papers that we missed out in the related work.

**Reviewer 2** raises a point about the baselines potentially not being anchored with previous numbers. Whilst it's true that we ran all of the baselines ourselves, we nevertheless used the author's publicly released code for all of the baselines, except HER. For HER, we used the OpenAI Baselines implementation. If we compare with Plappert et al.[1], they show results for FetchPush and FetchPickAndPlace in their Figure 3 (note 1 epoch = 95000 environment interactions). Although perhaps not immediately obvious, if you carefully compare our results with theirs using this scaling you should see that they (at least approximately) match up. To address some of reviewer 2's other questions: we agree that it would be an interesting line of work to explore similar approaches for more general environments. The reasons we focused on sparse reward, goal-conditioned environments is that it felt natural to combine the training of the GANs with hindsight experience relabelling (which is really at the core of our approach), as well as the fact that typical model-based approaches are likely to be inappropriate for solving these kinds of task. Regarding the comment about l199, it's true that other strategies can be used here. In our initial experiments we did try simply using a single future goal for the whole unrolled trajectory, but found this did not perform quite as well. At the very least this should have been mentioned, and so we will include a comment and perhaps consider adding an ablation demonstrating this. We appreciate the comments re: l195, l217 and l225 and agree that your suggested terminology would be clearer. Re: the comment about l227, we think this is a slight misunderstanding and something we need to make clearer in the text. All GANs in the ensemble are trained from batches taken from the same replay buffer. For each step of each imaginary rollout we choose a random GAN in the ensemble, however these imaginary rollouts are not stored in the buffer (which only stores the actual rollouts with the final actions chosen by the planner, along with some initial random trajectories). This is related to a comment by **reviewer 3** who asks for clarification about this as well as asking how important "mixing" GANs in this way was. Essentially, this is the primary way in which the ensemble (rather than just using a single GAN) was made use of, so in that sense the ablation in Appendix B demonstrates that it does have a somewhat significant effect. However, it is true that there are other ways we could have made use of the ensemble, but we have not considered these so far. Re: the comment about l235, $R$ is a typo (thank you for spotting this). We weren't 100% sure what you meant by "...(not an integer, depends on ordering...)", but just to clarify (and we will adopt this notation in the next draft): the planning process gives a score, $n_i$ to each of the Q initial seed actions $a_i$ (note the actions are continuous). We then define weights $w_i = e^{\alpha n_i}$ and return $a_t = \frac{\sum_{i=1}^{Q} w_i a_i}{\sum_{j=1}^{Q} w_j}$.

To address some of **reviewer 3's** other questions: the training curves do indeed account for the initial random trajectories (including those we discard and don't store in the replay buffer). M in figure 1 is supposed to represent the one-step predictive model, although we realise now there is an error in this (M should have both $s_t$ and $a_t$ as inputs) — we will fix this. We did consider removing the OSM regularization, but decided not to as we felt it was still an interesting experiment, despite the fairly small difference it made (having said this, the difference on FPAP was $\sim 5 - 10\%$, which looks small on the plot but is not entirely insignificant). Given that there was not much difference between noOSM and OSM, it's also the case that there is not any strong dependence on $\lambda$.

Addressing some of **reviewer 4's** comments: it's true that we use more updates than HER when training our model-based approach. However, previous work has shown increasing $num\_updates$ can actually degrade HER's performance slightly. We ran some experiments on FPAP to demonstrate this (and could add these to an appendix perhaps) — see the plot on the right. It's an interesting question to ask whether HER could also be improved by using an ensemble — however exactly *how* you would best make use of such an ensemble of policies/value functions in that context is not entirely trivial and an interesting research question in itself. We feel this is also true of employing HER with a forward model (again, a very interesting suggestion) — would you train DDPG+HER within a learned world model or use some other kind of model-based search to gather rollouts? How often would you update the model (or continuously update)? There are a number of considerations that, whilst interesting, go beyond a very straightforward baseline to compare with, in our opinion.

## Footnotes

[1] Multi-Goal Reinforcement Learning: Challenging Robotics Environments and Request for Research, 2018


[Meta-Review · NeurIPS 2020]

This paper proposes using an ensemble of GANs to learn a goal-conditioned forward model of trajectories for use in planning. The model is trained using a variant of hindsight experience replay, resulting in an agent that can succeed at sparse goal-conditioned tasks with much better data efficiency than model-free approaches. All reviewers highlighted the impressiveness of the experimental results, with R1 and R2 finding the approach very interesting, and R3 and R4 indicating the potential impact and interest this work will have. I agree that this paper will likely be of broad interest to the RL community at NeurIPS and therefore recommend acceptance. However, several reviewers also noted the lack of comparison to other model-based approaches. While PlanGAN is an interesting idea worth being published on its own, I agree the paper would be greatly strengthened by a comparison to an existing non-goal conditioned planning method, and I would like to see this addition in the camera-ready. Additionally, it’s not entirely clear to me what are the tradeoffs of modeling trajectories via a GAN rather than combining a goal-conditioned policy with a non-goal-conditioned state transition model (i.e., s_{t+1} = f(s_t, a_t); a_t = \pi(s_t, g)). For example, is it that the GAN is more robust to compounding model errors? Or is it that it’s easier to jointly model states and actions than the two separately? Indeed, R2 was also concerned that some of the choices made in the paper seem arbitrary or unjustified, and so I would therefore like to see more discussion of these considerations included.